# Parasitism of *Hirsutella rhossiliensis* on Different Nematodes and Its Endophytism Promoting Plant Growth and Resistance against Root-Knot Nematodes

**DOI:** 10.3390/jof10010068

**Published:** 2024-01-15

**Authors:** Xin Sun, Jiaqian Liao, Junru Lu, Runmao Lin, Manling Zou, Bingyan Xie, Xinyue Cheng

**Affiliations:** 1College of Life Sciences, Beijing Normal University, Beijing 100875, China; 2State Key Laboratory of Vegetable Biobreeding, Institute of Vegetables and Flowers, Chinese Academy of Agricultural Sciences, Beijing 100081, China; 3College of Plant Protection, Hainan University, Haikou 570228, China; 4Ministry of Education Key Laboratory for Biodiversity Science and Ecological Engineering, Beijing 100875, China

**Keywords:** nematophagous fungus, endoparasite, *Hirsutella rhossiliensis*, plant-parasitic nematode, parasitism, endophytism, plant growth promotion, plant resistance improvement, *Bursaphelenchus xylophilus*, green fluorescent protein (GFP)-tagged *transformant*

## Abstract

The endoparasitic fungus *Hirsutella rhossiliensis* is an important biocontrol agent of cyst nematodes in nature. To determine the potential parasitism of the fungus on a non-natural host, the pinewood nematode (*Bursaphelenchus xylophilus*) living in pine trees and the endophytic ability of the fungus on plants, in this paper, we first constructed and utilized a green fluorescent protein (GFP)-tagged *H. rhossiliensis* HR02 transformant to observe the fungal infection process on *B*. *xylophilus* and its colonization on *Arabidopsis* roots. Then, we compared the fungal parasitism on three species of nematodes with different lifestyles, and we found that the fungal parasitism is correlated with nematode species and stages. The parasitic effect of *H. rhossiliensis* on adults of *B. xylophilus* is similar to that on second-stage juveniles (J2) of the root-knot nematode *Meloidogyne incognita* after 24 h of inoculation, although the virulence of the fungus to second-stage juveniles of *M. incognita* is stronger than that to those of *B. xylophilus* and *Caenorhabditis elegans*. Moreover, the endophytism of *H. rhossiliensis* was confirmed. By applying an appropriate concentration of *H. rhossiliensis* conidial suspension (5 × 10^6^ spores/mL) in rhizosphere soil, it was found that the endophytic fungus can promote *A. thaliana* growth and reproduction, as well as improve host resistance against *M. incognita.* Our results provide a deeper understanding of the fungus *H. rhossiliensis* as a promising biocontrol agent against plant-parasitic nematodes.

## 1. Introduction

*Hirsutella rhossiliensis* Minter & Brady (synonym: *Hisutella heteroderae* Sturhan & Schneider) is an important endoparasitic fungus of nematodes, which was originally isolated from soil but later on the nematode *Heterodera humuli* Filipjev in Germany in 1980 [1]. The fungus is a dominant parasite of second-stage juveniles (J2) of the soybean cyst nematode (SCN) *H. glycines* in the farmland. It can be used as a biocontrol agent for controlling the plant-parasitic nematode [2,3,4]. The fungus produces conidia that adhere to the nematode cuticle and then form germ tubes to penetrate the nematode. After consuming and eventually killing the nematode, the fungus grows from the cadaver and produces new conidia for the next infection cycle [5]. Field observations and laboratory experiments have indicated that *H. rhossiliensis* may parasitize nematodes in a density-dependent manner [6]. So far, the fungus has been isolated from various nematode species worldwide and has a broad host range [2,3,7,8], including species of plant-parasitic nematodes, such as *Heterodera*, *Meloidogyne*, *Pratylenchus*, *Ditylenchus, Aphelenchoides* [1], *Criconemella* [9], *Anguina* [10] and *Xiphinema* [11]; species of entomopathogenic nematodes, such as *Steinernema* and *Heterorhabditis* [12]; as well as species of bacteria-feeding nematodes, such as *Anaplectus* and *Cephalobus* [1]. This indicates that the fungus can parasitize nematode genera with different trophic types. However, the parasitic effects of *H. rhossiliensis* on nematodes differ among the fungal isolates and the nematode species [13]. A parasitism assay showed that percentages of second-stage juveniles of *H. glycines* and *H. avenae* parasitized by the fungus *H. rhossiliensis* (mean 39.5% and 32.2%, respectively) were significantly different among the fungal isolates, even though those isolates were from the same host *H. glycines* [8]. Most *H. rhossiliensis* strains had a higher parasitism on *H. glycines* than on the root-knot nematode *Meloidogyne hapla*, and only a few isolates parasitized over 30% of *M. hapla* J2 [8]. It was also reported that *H. rhossiliensis* killed *Ditylenchus dipsaci* within four days, but only killed 45–65% of *Aphelenchoides fragariae* in that length of time. However, it killed *M. incognita* J2 in about two days [14]. Moreover, the fungal parasitism is correlated with the nematode stages. It was found that the mortality of *Caenorhabditis elegans* parasitized by *H. rhossiliensis* was negatively correlated with the worm size, i.e., the parasitism was higher for the younger stages. The parasitism of *C. elegans* L1 was correlated with that of *H. glycines* J2 [13]. Currently, parasitism assays of *H. rhossiliensis* are mainly carried out with soil-borne nematodes, and few are carried out with nematodes living above ground. A study reported that the parasitism of some *H. rhossiliensis* isolates was weak on the pinewood nematode *Bursaphelenchus xylophilus*, which lives in pine trees. The parasitism on *B. xylophilus* was similar to that on the northern root-knot nematode *M. hapla* and the entomopathogenic nematodes *H. bacteriophora* and *S. carpocapsae* [8]. So far, the parasitism of *H. rhossiliensis* on plant-parasitic nematodes living above ground has rarely been estimated. 

It is documented that most nematophagous fungi can endophytically colonize plant roots, such as the egg-parasite *Pochonia chlamydosporia* and the nematode-trapper *Arthrobotrys oligospora* [15,16,17,18]. Root colonization by nematophagous fungi can elicit plant defense responses and promote plant growth [19,20,21]. However, the endophytic ability of *H. rhossiliensis* is undetermined. It was once reported that, unlike the above two nematophagous fungi (*A. oligospora* and *P. chlamydosporia*), *H. rhossiliensis* could not seem to colonize barley roots endophytically [16,17]. By using light microscopy, it was observed that cortex and epidermal cells were free from hyphal colonization after the inoculation of *H. rhossiliensis* for three weeks. But the fungus seemed to colonize the rhizoplane abundantly, where it formed viable conidiophores [16,17]. Currently, labeling and tracing with fluorescent proteins are effective and reliable for studying the in vivo physiological activities of the fungal localization, adhesion and colonization [22]. Green fluorescent protein (GFP) was also successfully expressed in *P. chlamydosporia* [23]. The colonization of GFP-tagged *P. chlamydosporia* was observed in plant root [21,24,25]. These studies provide new methods for determining whether the endoparasite *H. rhossiliensis* lacks endophytic ability.

For visualizing the fungal infection and colonization in nematodes and plants, in this study, we first constructed fluorescent-protein-labeled transformants of *H. rhossiliensis* HR02, and then observed the fungal infection on nematodes and colonization in *Arabidopsis* roots by using fluorescence and confocal microscopy. In view of the fact that parasitic effects of *H. rhossiliensis* on nematodes differ among the fungal isolates, nematode species and development stages, we tested and compared the parasitism of *H. rhossiliensis* HR02 on different nematodes (*B. xylophilus*, *M. incognita* and *C. elegans*) and in different stages. Meanwhile, we also estimated roles of the fungus in soil to promote plant growth and resistance to plant-parasitic nematodes.

## 2. Material and Methods

### 2.1. Microbial Strains and Culture Conditions

The endoparasite *H. rhossiliensis* HR02, which was isolated from soybean cyst nematode (*H. glycines*) (a gift of Prof. X.Z. Liu, Institute of Microbiology, Chinese Academy of Sciences, Beijing, China), was cultured on potato dextrose agar (PDA) at 25 °C in an incubator. For harvesting conidial spores, the fungus was cultured for two weeks. Then, the colonies were washed with sterilized distilled water (DW) and filtered with four layers of sterile filter paper. The spores were collected using a centrifuge at 10,000 rpm for 10 min at 4 °C. The spores were resuspended with sterilized DW (containing 0.1% Tween-20). The concentration of conidial suspension was determined with a hemocytometer. The fungus *Botrytis cinerea* was cultured on PDA at 25 °C in the dark and used for culturing the nematode *B. xylophilus*. The bacterium *Escherichia coli* strain OP50 (Bioscibio, Hangzhou, China) was cultured on nematode growth medium (NGM) [26] at 37 °C and used for feeding the nematode *C. elegans*.

### 2.2. Nematodes

Three nematodes with different lifestyles were used for fungal parasitism assay. The pinewood nematode *B. xylophilus* strain ZJSS, which was isolated from *Pinus massoniana* and deposited in our lab (BNU, Beijing, China), was cultured on the fungal mat of *B. cinerea* grown on PDA at 25 °C. The cultured nematodes were washed from the plates using sterilized DW and used for fungal inoculation immediately. Worms of different stages can be distinguished, based on the nematode length and morphology [27]. The southern root-knot nematode *M. incognita*, which was isolated from tomato plant (*Solanum lycopersicum*) and kept in our lab (IVF, CAAS, Beijing, China) after morphology and molecular identification, was multiplied on the water spinach (*Ipomoea aquatica*). Three primer pairs were used for molecular identification, including rDNA-F/R (TTGATTACGTCCCTGCCCTTT/TTTCACTCGCCGTTACTAAGG) [28], NAD5-F/R (TATTTTTTGTTTGAGATATATTAG/CGTGAATCTTGATTTTCCATTTTT) [29] and Inc-K14-F/R (GGGATGTGTAAATGCTCCTG/CCCGCTACACCCTCAACTTC) [30]. Egg masses of *M. incognita* collected from roots were treated with 3% NaClO for 3 min and then followed by rinsing with sterile DW. The eggs were incubated in sterilized DW at room temperature (22–25 °C). After 2–3 days, J2 were hatched and collected for inoculation. The free-living nematode *C. elegans* Bristol N2 strain (a gift of Prof. X. Liu, the Capital Normal University, Beijing, China) was cultured on NGM plates seeded with *Escherichia coli* strain OP50 at 21 °C [26]. Worms of each stage were harvested by synchronizing culture. First, eggs were collected with M9 buffer and then planted on NGM plates. After culture at 21 °C for 9, 21, 29, 38, 56 h, worms of L1, L2, L3, L4 and adult stages were collected, respectively. The development time followed those of previous studies [31,32]. Harvested worms were used for fungus inoculation immediately. 

### 2.3. Arabidopsis thaliana

The plant *A. thaliana* ecotype Col-0 (from the Nottingham Arabidopsis Stock Centre, NASC) was cultured and used for fungal endophytism assay. Surface-sterilized *A. thaliana* seeds were sown on Murashige–Skoog (MS) medium and cultured vertically in an incubator with a 16 h light/8 h dark photoperiod for 10 days. Then, the plantlets were transplanted into pots containing a mixture of peat soil (KLASMANN876, Geeste, Germany), perlite and vermiculite (2:1:1) and grown at 21 °C with a 16 h light/8 h dark photoperiod. 

### 2.4. Genetic Transformation of H. rhossiliensis with Green and Red Fluorescent Proteins

Two expression vectors (pCH-sGFP and pGNT-mCherry), which were previously constructed [33] and deposited in our lab (IVF, CAAS, Beijing, China), were used for the fungal transformation. The pCH-sGFP vector contains a hygromycin antibiotic resistance gene and the pGNT-mCherry vector contains a geneticin antibiotic resistance gene, which were used as selective markers for transformant screening. The fluorescent protein genes can be expressed under the control of the *gpdA* promotor and *trpC* terminator. The PEG-mediated protoplast transformation method was used for *H. rhossiliensis* genetic transformation. The preparation of *H. rhossiliensis* protoplasts was performed according to the description in a previous study [34]. Briefly, freshly cultured conidial spores were grown in potato dextrose broth (PDB) at 25 °C for 72 h with shaking at 180 rpm. Then, 1 g freshly germinated conidial pellets were collected and digested with Yatalase enzyme (20 mg/mL) (Takara, Dalian, China) at 28 °C for 3–4 h. After filtering, rinsing repeatedly with STC buffer (0.7 M sorbitol, 50 mM CaCl_2_·2H_2_O, 10 mM Tris/HCl, pH 7.5) and centrifuging at 3000 rpm for 15 min at 4 °C, the protoplasts were collected. The protoplast deposit was resuspended with STC buffer and the concentration was adjusted to 2 × 10^8^ spores/mL. The prepared protoplasts were directly used for subsequent transformation. 

Then, PEG-mediated protoplast transformation was performed. About 5 µg plasmid (pCH-sGFP or pGNT-mCherry) was added to 10 µL aurintricarboxylic acid (100 mM) and quantified to 60 µL with TEC buffer and then incubated on ice for 20 min. After centrifugation at 12,000 rpm for 2 min at 4 °C, the supernatant was added to 100 µL protoplasts and incubated on ice. Then, an equal volume of PEG was added to the protoplast mixture and incubated at room temperature. An amount of 1 mL STC buffer was added to the mixture. After centrifugation at 3000 rpm for 5 min, the protoplasts were isolated and then resuspended with STC buffer. Adding 50 µL transformed protoplasts to 10 mL minimal medium (MM), the mixture was coated on MM agar plates and cultured at 25 °C after solidification. After 12–20 h, 5 mL T-top containing 200 μg/mL hygromycin was laid on the transformation plates and cultured upside-down at 25 °C until hyphae grew on the surface of T-top. Then, colonies were picked up and each was cultured on a new PDA plate containing 200 μg/mL hygromycin/geneticin. 

Positive transformants were first verified by PCR detection. For GFP-tagged transformants, a fragment of *hph* gene was amplified with the primer pair hph-f/-r (hph-f 5′-GAGCCTGACCTATTGCATCTC-3′, hph-r 5′-CCGTCAACCAAGCTCTGATAG-3′). For mCherry-tagged transformants, a fragment of geneticin resistance gene was amplified with primer pair g418-f/-r (g418-f 5′-GAGGCTATTCGGCTATGACTGG-3′, g418-r 5′-TCCGCCACACCCAGCCGGC-3′). Both genes were present in the expression vectors pCH-sGFP and pGNT-mCherry, respectively. Then, mycelia of the positive candidates were further observed using a Zeiss fluorescent microscope (Axio Scope A1, Carl Zeiss, Suzhou, China), with an Axiocam 506 color camera and fluorescence lamp illuminations (X-Cite 120Q) to confirm the expression of the fluorescent proteins. 

### 2.5. Observation of the Fungal Infection Process on Nematodes

We used a GFP-tagged *H. rhossiliensis* transformant (HR-eGFP1) for infection observation. The fungus was cultured on PDA at 25 °C for two weeks. Then, about 300 freshly cultured nematodes of *B. xylophilus* (including worms of J2 to J4 and adults), *C. elegans* (including worms of L1 to L4 and adults) and *M. incognita* (J2) were inoculated on the fungal mat of each plate, respectively (Appendix A). After 2–4 h inoculation, these nematodes were collected and transferred to a 90 mm glass dish for observation. Nematodes with conidial spores attached were picked out for further observation. The slides were prepared as described in a previous study [35]. One drop of 4% boiled agarose was placed on the center of a sterilized slide, and a sterilized coverslip was placed onto the agarose drop immediately and pressed down to spread the agarose. Once the agarose was solidified, the coverslip was removed. Nematodes with conidial spores attached were picked up and transferred to the centers of the agarose pads, and then they were covered with coverslips and sealed with Vaseline. The slides were kept in a container at 25 °C. The infection process was observed and photographed with a ZEISS fluorescence microscope (Zeiss Axio Scope. A1, Suzhou, China). 

### 2.6. Parasitism Assay of H. rhossiliensis HR02 on Nematodes

Three nematodes (*B. xylophilus*, *M. incognita* and *C. elegans*) were used for the fungal parasitism assay. A 10 μL conidial suspension (approx. 2 × 10^7^ spores/mL) of *H. rhossiliensis* HR02 was spread onto the surface of each well of 12-well tissue culture plates, which were filled with 2 mL PDA in each well, and then cultured at 25 °C for two weeks. An amount of 20 μL of nematode suspension (approx. 300 freshly cultured worms in 4.5 mM KCl solution) was added to the center of each well, according to the method in a previous study [36]. The second-stage juveniles/larvae (J2/L2) of the three nematodes were used for the test. After 2, 8, 16 and 24 h of incubation, the nematodes were washed with sterilized DW containing 0.1% Tween-20. About 100 individuals from each well were randomly picked out for observation. Worms attached with fungal spores or colonized by the fungus were determined using an inverted microscope (Olympus, Tokyo, Japan) with 20× objective lens. Percentage of fungal parasitism on each nematode was calculated. Moreover, we compared parasitism and mortality of *B. xylophilus* in different stages (J2, J3/J4 and adults) after 24, 48, 72, 96 and 120 h of inoculation. Mortality of the same-stage worms with no fungal inoculation was taken as control. Corrected mortality was calculated with the following formula: Corrected mortality (%) = (treated mortality % − control mortality %)/(1 − control mortality %) × 100%. We also compared parasitism of *C. elegans* in different stages (L1 to L4, and adults) after 24 h of inoculation. All experiments were repeated three times, and each treatment had three replicates each time. 

### 2.7. Observation of H. rhossiliensis Colonizing Arabidopsis Roots

An *H. rhossiliensis*-GFP-tagged transformant (HR-eGFP1) was cultured for two weeks on PDA medium. Then, conidial spores were collected by washing with sterilized DW containing 0.1% Tween-20 and the suspension was adjusted to a concentration of 2 × 10^8^ spores/mL. The roots of *A. thaliana* cultured for 2–3 weeks on MS medium were dipped into the conidial suspension for 48 h at 25 °C with 60 rpm of shaking. Then, the roots were rinsed with sterilized DW to remove the spores on the root surface. The cleaned roots were placed on slides and covered with coverslip. Observation and photographing were performed with a confocal microscopy (ZEISS, LSM 880, Jena, Germany). 

### 2.8. Measurement of Fungal Effects on A. thaliana Growth and Resistance to Root-Knot Nematodes

We measured and compared the growth parameters of *Arabidopsis* plants inoculated with different concentrations of *H. rhossiliensis* conidial suspension. *H. rhossiliensis* conidial suspension was adjusted to different concentrations in sterilized DW (approx. 5 × 10^3^, 5 × 10^6^, 5 × 10^9^ spores/mL). After *A. thaliana* seedlings were transplanted into pots containing a mixture of peat soil (KLASMANN876, Geeste, Germany), perlite and vermiculite (2:1:1) for five days, a 10 mL conidial suspension was poured into the rhizosphere soil of each plant. Another five days later, an additional 10 mL conidial suspension was poured into the rhizosphere soil of each plant. No fungal inoculation was performed on control plants. All plants were randomly placed in the culture chamber and grown at 21 °C with a 16 h light/8 h dark photoperiod. Each treatment had 20 pot-grown *Arabidopsis* plants, and the experiment was repeated three times. Plant height was measured after 10, 40, 60 days of the first inoculation. Meanwhile, at the last time (60 days of inoculation), root weight, number of fruit pods and weight of seeds were measured. The first flowering time of each plant was recorded. For each parameter, the mean size of each treatment was calculated and compared among treatments. 

Moreover, we evaluated the effects of fungal endophytism on host resistance against root-knot nematodes. Plants were planted as above and the fungal conidia were inoculated with a suspension of 5 × 10^6^ spores/mL (20 mL), which was the optimum concentration for plant growth. Plants with no fungal inoculation were taken as the control. Two weeks after *Arabidopsis* plants were transplanted, 200–300 individuals of *M. incognita* J2 were inoculated on the rhizosphere soil of each plant. After 45 days of nematode inoculation, the numbers of root galls, egg mass and eggs per mass on each plant were measured. Each treatment had 20 pot-grown *Arabidopsis* plants, and the experiment was repeated three times.

### 2.9. Data Analysis

Statistical analysis was performed for comparison of the mean parasitism among nematodes and stages, as well as comparison of the mean sizes of each growth parameter among different treatments of the above experiments. We first performed Levene’s test for the homogeneity of variance across groups and the Shapiro–Wilk normality test for the Gaussian distribution, using the car package in R. Then, we used analysis of variance (ANOVA) for the comparison of fungal parasitism, nematode mortality and plant growth parameters. ANOVA was conducted (F-test and Tukey test) via the Anova function in the car package [37]. *T*-test was used for comparison of host resistance to *M. incognita*. All data analyses and visualizations were performed in R 4.3.0 (https://www.r-project.org, accessed on 21 April 2023).

## 3. Results

### 3.1. Acquisition of GFP-Tagged and mCherry-Tagged Transformants of H. rhossiliensis HR02

PCR amplification was performed to detect positive transformants. A fragment of the *hph* gene (587 bp) and a fragment of the geneticin resistance gene (597 bp) were, respectively, amplified from the GFP-tagged and mCherry-tagged transformants, indicating that the inserted T-DNA were integrated into the fungal genome of *H. rhossiliensis* HR02 (Figure 1a,b). Using the ZEISS fluorescence upright microscope (Zeiss Axio Scope A1, Suzhou, China), with filter sets of excitation at 460–480 nm and emission at 505–530 nm for GFP, and excitation at 540–552 nm and emission at 575–640 nm for mCherry, fluorescent signals were monitored in hyphae and conidia of these positive transformants (Figure 1c,d), but no signal in the wild-type strain (Figure 1e), indicating that the fluorescent proteins were expressed in those positive transformants. Observations of the colony, hyphae and conidia showed no obvious difference in morphology among the three *H. rhossiliensis* HR02 strains, i.e., the GFP-tagged transformant (Figure 1c), the mCherry-tagged transformant (Figure 1d) and the wild-type strain (Figure 1e), indicating no obvious influence on the fungus by insertion of the two exogenous genes into the genome of *H. rhossiliensis* HR02. Seeing that the green fluorescent signal (Figure 1c) is stronger than the red fluorescent signal (Figure 1d), we used a GFP-tagged transformant for infection observation.

### 3.2. Observation of the Fungal Infection Process on Nematodes

We used a GFP-tagged strain (HR-eGFP1) to observe the fungal infection process on the nematode *B. xylophilus* (Figure 2a–e). Using the fluorescence microscope, it is clearly observed that conidial spores are attached to the cuticle of the nematode (Figure 2a) and form germination tubes (Figure 2b). Notably, more spores are gathered at the worm head. Then, the fungal hyphae penetrate into the nematode body cavity and grow in the nematode body (Figure 2c,d). Finally, nutrients of the nematode are exhausted and the fungal hyphae grow out from the colonized nematode and sporulate for the next invasion (Figure 2e). A similar process was also observed on the other two nematodes. More fungal spores are gathered at the head and tail of *M. incognita* (Appendix A) and finally grow out of the nematode body after nutrients are used up (Appendix A). Moreover, we observed that the fungal spores could also enter the intestines of *C. elegans* by swallowing through the buccal cavity, and a similar infection process may take place in the nematode’s intestine (Appendix A). 

### 3.3. Comparison of the Fungal Parasitism on Different Nematodes

We compared the parasitism rates of the fungus *H. rhossiliensis* HR02 on the second- stage juveniles/larvae (J2/L2) of the three nematodes (*M. incognita*, *B. xylophilus* and *C. elegans*). The result shows that the fungal parasitism rates on the three nematodes are obviously different, although the differences are not statistically significant in the first two hours after inoculation (Figure 3a). After eight hours of inoculation, the fungal parasitism on *M. incognita* (mean 70.5%) is significantly higher than those on *B. xylophilus* (mean 26.4%) and *C. elegans* (mean 27.6%). After 16 h of inoculation, more than 90% of worms of *M. incognita* are parasitized, but only about 31.7% of *B. xylophilus* and 43% of *C. elegans* are parasitized. After 24 h of inoculation, more than 50% worms are parasitized in both *B. xylophilus* (52.6%) and *C. elegans* (59.5%) (Figure 3a). At that time (24 hpi), the mortality of *M. incognita* is about 28%, but it is less than 3% in the other two nematodes. This result indicates that *H. rhossiliensis* HR02 has a stronger virulence to the second-stage juveniles of *M. incognita* than to those of *B. xylophilus* and *C. elegans*. 

Then, we tested the parasitism of the fungus in different stages of *B. xylophilus* and *C. elegans*. The result shows that, for *B. xylophilus*, the fungal parasitism increases with the worm stages, i.e., the parasitism on adults is obviously higher than that on juveniles, and the parasitism on J3/J4 is higher than that on J2 (Figure 3b). On the contrary, for *C. elegans*, the fungal parasitism decreases with the stages. After 24 h of infection, the fungal parasitism decreases from 62.8% on L1 to 48.6% on adults (Figure 3c). However, the pattern of infection-related mortality of *B. xylophilus* is different from the pattern of parasitism. On the first two days of infection, the mortality of adults is larger than that of juveniles. But afterwards, the mortality of J3/J4 is notably higher than that of adults and J2. After infection for five days, more than 90% of J3/J4 and about 65% of adults are dead, but less than 40% of J2 are dead at that time (Figure 3d). The results indicate that the fungal parasitism is obviously correlated with the nematode species and stages. 

### 3.4. Endophytic Colonization of H. rhossiliensis HR02 on Arabidopsis Roots

For determining whether *H. rhossiliensis* HR02 can colonize plants, we used conidia of HR-eGFP1 to infect the root of *A. thaliana* for 48 h, and we then monitored roots using a confocal microscope (ZEISS, LSM 880). Under a high-power objective lens, it is clearly observed that the fungal spores adhere to the root surface (Figure 4a), form germination tubes and penetrate into the epidermal cells (Figure 4b). Then, hyphae grow and ramify within the intercellular space of the root cortex (Figure 4c), and finally the crossed mycelium forms a web around the roots (Figure 4d). Our result indicates that the fungus *H. rhossiliensis* has an endophytic ability to colonize plant roots.

### 3.5. Effects of Endophytic H. rhossiliensis on Promoting Plant Growth

We measured and compared growth indexes of *A. thaliana* grown in soils applied with different concentrations (5 × 10^3^, 5 × 10^6^ and 5 × 10^9^ spores/mL) of *H. rhossiliensis* HR02 conidial suspension (in DW) in the rhizosphere, which include the height, root weight, first flowering time, number of fruit pods and weight of seeds. Plants grown in soil without fungal conidia application were taken as the control. The result shows that plant growth rates are different among groups treated with different concentrations of conidial suspension (Figure 5a). The average heights of plants treated with conidial suspensions are significantly higher than that of the control, except that of plants treated with the high conidial concentration (5 × 10^9^ spores/mL) for a long time (60 days, Figure 5b). Plants treated with the concentration of 5 × 10^6^ spores/mL grow fast, followed by those treated with the concentration of 5 × 10^3^ spores/mL. Within the first ten days of inoculation, the plant growth rates are similar among the three fungal treatments. Thereafter, the growth rate is slowed down in the group treated with the high concentration (5 × 10^9^ spores/mL), whose average height is obviously lower than those of the other two treatments, but still higher than that of the control. But after 60 days of inoculation, the average height of the treatment with the high concentration (mean 20.5 cm) is not only significantly lower than those of the treatments with 5 × 10^6^ spores/mL (mean 34.6 cm) and 5 × 10^3^ spores/mL (mean 32.4 cm), but also lower than that of the control (mean 26.3 cm, Figure 5b). Similar results are also observed on the root weights of *A. thaliana*. After 60 days of inoculation, the average root weight of plants treated with 5 × 10^6^ spores/mL (mean 96.7 mg) is obviously higher than those of plants with 5 × 10^3^ spores/mL (mean 32.8 mg), and both are obviously higher than that of the control (mean 20.9 mg). But the root weight of plants treated with 5 × 10^9^ spores/mL (mean 11.0 mg) is lower than those of all the above (Figure 5c). All differences are statistically significant (*p* < 0.01). Our results indicate that inoculation with moderate conidial concentrations of *H. rhossiliensis* can promote plant growth. However, inoculation with a high conidial concentration may decrease the improvement over time, and even inhibits plant growth. 

We also recorded the first flowering date of each plant treated with different concentrations of fungal conidia and compared their growth periods from vegetative to reproductive stage. It was found that the growth period of plants treated with the concentration of 5 × 10^3^ spores/mL (mean 29.9 days) is similar to that of plants treated with 5 × 10^6^ spores/mL (mean 29.4 days), with both obviously shorter than that of the control (mean 32.4 days) and even shorter than that of the plants with the concentration of 5 × 10^9^ spores/mL (mean 35.9 days) (Figure 5d). The comparison of fruits shows that, after 60 days of inoculation, the plants treated with the concentration of 5 × 10^6^ spores/mL can produce more seed pods (mean 138 pots/plant) with heavier seed weights (mean 76 mg/plant), compared to the plants treated with 5 × 10^3^ spores/mL (mean 111 pots/plant, 30.9 mg/plant). Both are obviously greater in number and heavier than those of the control (mean 64 pots/plant, 23.6 mg/plant). Plants treated with the concentration of 5 × 10^9^ spores/mL produce fewer seed pots with lighter seed weights (mean 53 pots/plant, 7.6 mg/plant) (Figure 5e,f). All differences among treatments are statistically significant. 

In summary, the application of moderate conidial concentrations (5 × 10^6^ spores/mL) of *H. rhossiliensis* HR02 in rhizosphere soil can increase plant height, root weight, number of seed pots and seed weight, as well as advancing the flowering period (Figure 5). The results indicate that the endophytism of *H. rhossiliensis* can promote plant growth and reproduction. 

### 3.6. Effects of Endophytic H. rhossiliensis on Improving Plant Resistance to Root-Knot Nematodes

We tested the endophytic effects of *H. rhossiliensis* HR02 on host resistance against the root-knot nematode *M. incognita*. After 45 days of nematode inoculation on roots of *A. thaliana* grown in soil treated with or without conidial suspension (5 × 10^6^ spores/mL, 20 mL), the numbers of root galls, egg masses and eggs per mass on each plant were counted. By comparison, it was shown that the number of galls on roots of plants growing in soil inoculated with conidia (mean 10 galls/plant, Figure 6a) is obviously less than that of the control (mean 25 galls/plant, Figure 6b). The difference between the two treatments is statistically significant (*df* = 118, *t* = −9.944, *p* < 0.001; Figure 6c). The number of egg masses in the fungus treatment group (mean 2 masses/plant) is also significantly less than that of the control (mean 8 masses/plant; *df* = 83.6, *t* = −7.9712, *p* < 0.001; Figure 6d). The number of eggs per mass on each plant in the fungus treatment group (mean 41 eggs/mass/plant) is also significantly less than that of the control (mean 92 eggs/mass/plant; *df* = 97, *t* = −21.017, *p* < 0.001; Figure 6e). The results indicate that the endophytic *H. rhossiliensis* can improve plant resistance against plant-parasitic nematodes and alleviate plant damage.

## 4. Discussion

The endoparasitic fungus *H. rhossiliensi* is a promising biocontrol agent of plant-parasitic nematodes, especially to cyst nematodes [38]. Previous studies mainly focused on the fungal parasitism on soil-borne nematodes [8,9,13,39], but few related to nematodes living above ground. In this study, we tested the fungal parasitism in the pinewood nematode *B. xylophilus*, which lives in pine trees and is an important invasive nematode in Asia and Europe. Our result shows that *H. rhossiliensis* HR02 has an obvious effect on *B. xylophilus*. After 24 h of conidial inoculation on adults of *B. xylophilus*, its parasitism and mortality are about 88% and 33%, respectively (Figure 3b,c), which are close to those of the fungus on J2 of the root-knot nematode *M. incognita*, with about 94% of parasitism and 28% of mortality at the same time (Figure 3a). The mortality of *B. xylophilus* increases with the parasitic time, especially in J3/J4. After 72 h of inoculation, the mortality of J3/J4 (74%) is obviously higher than that of adults (52%) (Figure 3c). The parasitism and mortality of J2 are lower than those of J3/J4 and adults. However, the parasitic pattern of *H. rhossiliensi* on *C. elegans* is different, on which the fungal parasitism decreases with nematode stages, from 63% of L1 larvae to 49% of adults after 24 h of inoculation (Figure 3d). This result is similar to a previous report that the parasitism of *C. elegans* by *H. rhossiliensis* is negatively correlated with larva size, i.e., parasitism is higher for the younger stages [13]. The comparison of fungal parasitism on second-stage juveniles/larvae of different nematodes shows that the fungus *H. rhossiliensis* HR02 is much more virulent to *M. incognita* than to *B. xylophilus* and *C. elegans* (Figure 3a). We speculate that each species of nematodes perhaps has its specific defensive mechanisms and immune responses to pathogens in long-term evolution. Genomic and dualRNA-seq analyses may provide useful information to understand mechanisms of the fungus–nematode interaction and to explain diverse effects of the fungus on different nematodes and stages. 

For visualizing the process of the fungus infection on nematodes and colonization on plant roots, using the PEG-mediated protoplast transformation method, two fluorescent proteins (GFP, mCherry) were successfully expressed in the fungus *H. rhossiliensis* HR02 (Figure 1), indicating that the genetic transformation system of *H. rhossiliensis* is constructed. Then, we used a GFP-tagged strain HR-eGFP1 to observe the infection process with the aid of fluorescence microscopy or confocal microscopy. A similar infection process was observed on both nematodes and plant roots, i.e., conidial spore adhesion, germination tube formation and penetration into host cuticle/epidermis, as well as hypha growth in host tissue (Figure 2 and Figure 4). Endophytic colonization of the fungus *H. rhossiliensis* on *A. thaliana* roots is clearly observed (Figure 4). Our result indicates that the fungus *H. rhossiliensis* has an endophytic ability, identical to the egg-parasitic fungus *P. chIamydosporia* and the nematode-trapping fungus *A. oligospora*. The result is different from the previous report that *H. rhossiliensis* had a lack of endophytic ability to the plant by observation with light microscopy [16,17]. So far, gene families associated with endophytism in the *Arabidopsis* root mycobiome have been reported [40]. The genetic determinants of *H. rhossiliensis* endophytism are revealed along with the fungal genome sequencing. Moreover, the endophytic effects of the fungus *H. rhossiliensis* on plant growth and defense against plant-parasitic nematodes are also confirmed in our study. By comparison of *A. thaliana* grown in soil applied with different concentrations of conidial suspensions of *H*. *rhossiliensis* HR02 in the rhizosphere, it is found that inoculation of an appropriate conidial concentration (5 × 10^6^ spores/mL) can significantly promote plant growth, advance the transition from vegetative to reproductive growth, increase seed pods and seed weight (Figure 5), as well as improve plant resistance to plant-parasitic nematodes (Figure 6). However, inoculation of a high conidial concentration (5 × 10^9^ spores/mL) over a long time (60 days) has negative effects on plants (Figure 5). Currently, some of the mechanisms involved in promoting plant growth by means of endophytic fungi are known, including the increasing access to nutrients, the production of plant hormones, the ethylene amount reduction or the increase in water acquisition [41]. But the mechanisms of *H*. *rhossiliensis* promoting plant growth and resistance against plant-parasitic nematodes are still unknown and will be investigated in a future study. 

## Figures and Tables

**Figure 1 jof-10-00068-f001:**
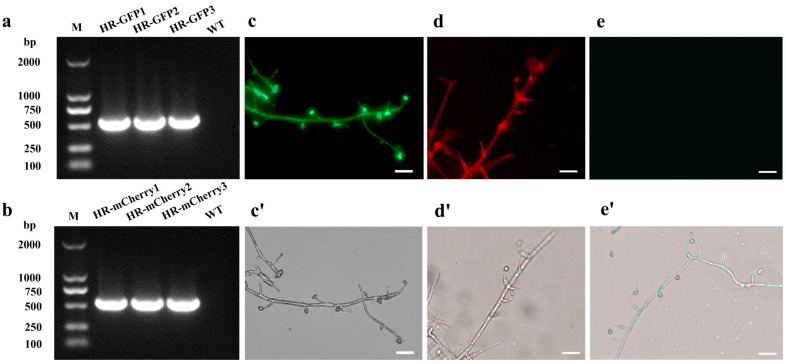
Verification of positive transformants of *H. rhossiliensis* HR02 by PCR detection and fluorescence survey. PCR amplification of *hph* gene (587 bp) in GFP-tagged transformants (**a**) and of the geneticin resistance gene (597 bp) in mCherry-tagged transformants (**b**). The hygromycin antibiotic resistance gene and the geneticin resistance gene are present in the pCH-sGFP and the pGNT-mCherry expression vectors, respectively. Hypha and conidia of GFP-tagged transformant (**c**,**c′**), of mCherry-tagged transformant (**d**,**d′**) and of the wild-type strain (**e**,**e′**) imaged using a fluorescence microscope, with fluorescence condition (**c**–**e**) and bright field (**c′**–**e′**), with 20× objective lens and 10× ocular lens. Scale bar = 5 μm.

**Figure 2 jof-10-00068-f002:**
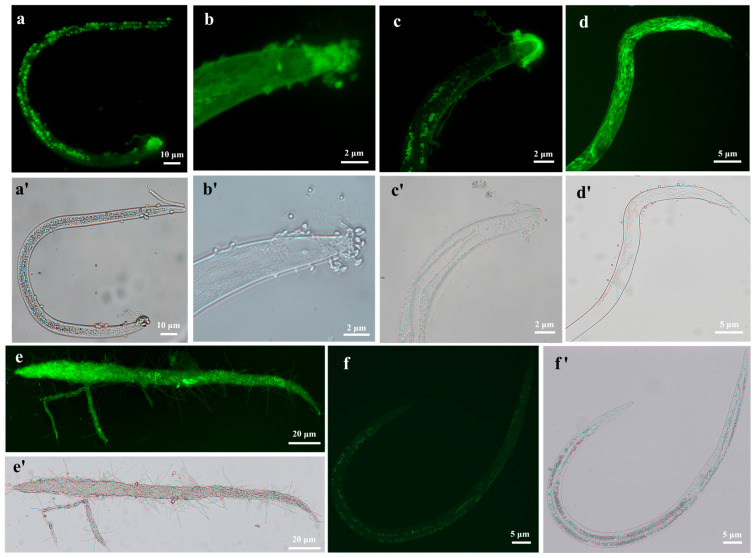
Visualization of the fungal infection on *B. xylophilus* by using a GFP-tagged transformant of *H. rhossiliensis* HR02 (HR-eGFP1). Images were taken using a ZEISS fluorescence microscope, with fluorescence condition (**a**–**e**) and bright field (**a′**–**e′**). Conidial spores are attached to the cuticle and gathered at the head of the nematode (**a**,**b**); germination tubes form and penetrate into the nematode body (**b**,**c**); fungal hyphae grow in the nematode body and consume the contents of the nematode (**d**); finally, the fungal hyphae grow out from the nematode (**e**). Weak autofluorescence is observed in the body of the control (uninfected nematode) (**f**), and image with bright field (**f’**). Imaged worms are J3/J4 (**a**–**d**,**f**) and adult (**e**). The bars marked with 2, 5, 10, 20 µm indicate images under 40×, 20×, 10× and 5× objective lenses, respectively.

**Figure 3 jof-10-00068-f003:**
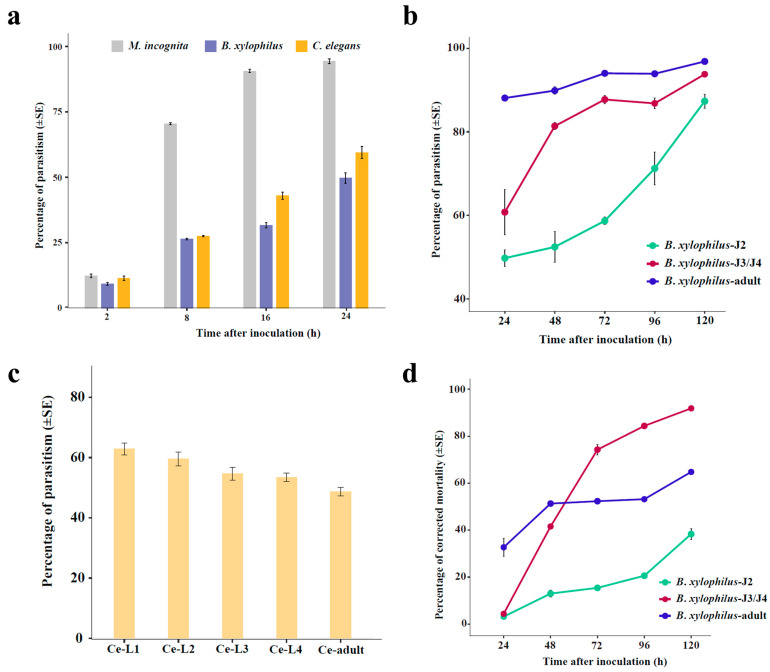
Comparison of fungal parasitism on different species and stages of nematodes. Parasitism of the fungus *H. rhossiliensis* HR02 on second-stage juveniles/larvae (J2/L2) of the three nematode species (*M. incognita*, *B. xylophilus* and *C. elegans*) (**a**). Parasitic dynamics of the fungus on *B. xylophilus* in different stages (J2, J3/J4, adult) (**b**). Fungal parasitism on *C. elegans* in different stages after 24 h of inoculation (**c**). Corrected mortality of *B. xylophilus* in different stages (J2, J3/J4, adult) (**d**). The bars represent standard errors.

**Figure 4 jof-10-00068-f004:**
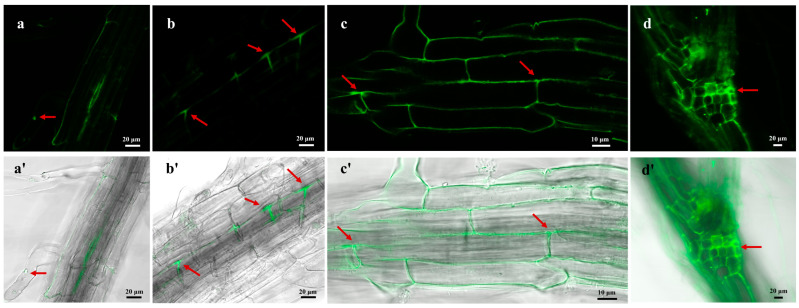
Visualization of colonization process of *H. rhossiliensis* HR02 (HR-eGFP1) on *A. thaliana* roots. Fluorescence images in the upper row (**a**–**d**) and composite images (bright field + fluorescence) in the lower row (**a′**–**d′**) were taken by confocal microscopy. The fungal spores adhere to the root surface (**a**); germination tubes form and hyphae penetrate into the epidermal cells (**b**); hyphae grow and ramify within the intercellular space of root cortex (**c**); mycelia continue to grow and further form a web along the intercellular space of the plant root (**d**). Arrows indicate the status of conidia adhering, penetration, growth and ramification on the root (**a**–**d**). Images from left to right are under 40×, 40×, 63× and 20× objective lenses, respectively.

**Figure 5 jof-10-00068-f005:**
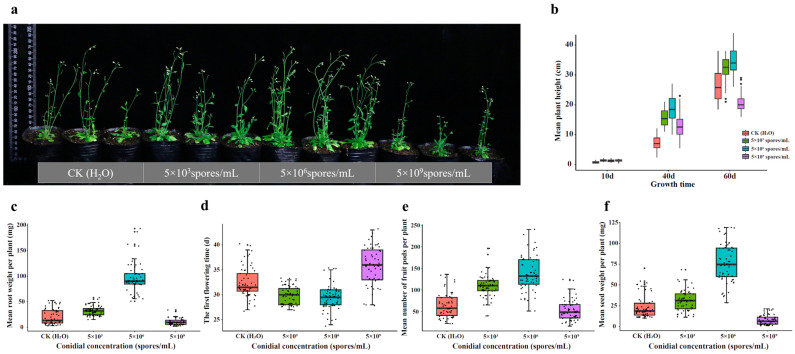
Comparison of growth indexes of *A. thaliana* grown in soils treated with different conidial concentrations of *H. rhossiliensis* HR02. Plant growth status (**a**), plant heights at different times (**b**), root weight (**c**), first flowering time (**d**), number of seed pots (**e**) and seed weight (**f**).

**Figure 6 jof-10-00068-f006:**
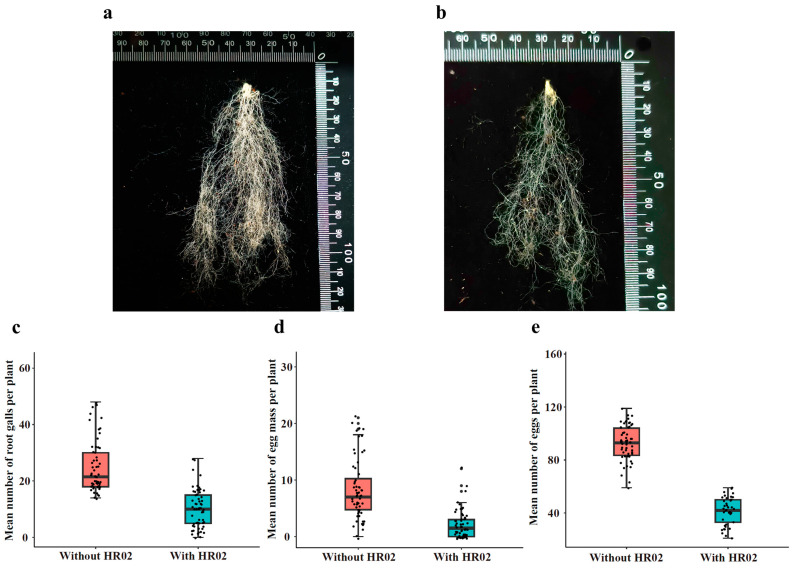
Effects of endophytic *H. rhossiliensis* on *A. thaliana* resistance to *M. incognita*. Roots of a plant grown in soil treated with (**a**) or without (**b**) conidial suspension of HR02 (5 × 10^6^ spores/mL). Number of root galls per plant (**c**), egg masses per plant (**d**) and eggs per mass on each plant (**e**). For the three indexes, differences between the two treatments were statistically significant (*p* < 0.001), determined using Student’s *t*-test.

## Data Availability

Data are contained within the article and Appendix A.

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
