# Peer review of "Parasitism of Hirsutella rhossiliensis on Different Nematodes and Its Endophytism Promoting Plant Growth and Resistance against Root-Knot Nematodes"

_jof, 2024, doi:10.3390/jof10010068_

Round 1
Reviewer 1 Report
Comments and Suggestions for Authors
Thanks for dealing with the previous suggestions/comments to the paper. Please revise the next comments/suggestions made to the new version of the manuscript (see Authors response for review1jof_Dec version) and PDFs.

Reviewer 2 Report
Comments and Suggestions for Authors
The authors examine experimentally the ability of the fungus Hirsutella rhossiliensis to infect and kill the non-natural host pinewood nematode (Bursaphelenchus xylophilus), and compare its parasitic efficiency with its respective ability to infect two other nematodes (Meloidogyne incognita and Caenorhabditis elegans). They additionally examine its ability to live endophytically (in Arabidopsis thalania) and thus, promote the plant's growth and its plant host resistance against nematodes.
This manuscript is an interesting revised version, after a first round with two reviewers who suggested how to improve the original version. This revised manuscript is improved, and the authors have addressed satisfactorily, almost all questions raised. However, in my opinion, they have to further clarify a couple of questions before this manuscript is suitable for publication in Journal of Fungi. In specific:
1. Iit is not clear (at least to me) in the last paragraph of the introduction what was their aim of this work. In other words, since there were other studies showing that H. rhossiliensis is neither a good parasite on B. xylophilus nor a good endophyte, it is not clear why the authors got into the trouble of researching the H. rhossiliensis competence as a pest of B. xylophilus or as an endophyte. I believe they have to elaborate on that.
2. It is not clear to me, why the authors had to have a second round of inoculation with conidia at the "rhizosphere soil of each plant" (l. 221-222). Was this second inoculation the starting point of counting time (10-60 days after inoculation)?
3. Why have the authors chosen to test parasitisim on different stages of C. elegans and B. xylophilus (l. 312-313) instead of B. xylophilus and M. incognita to which H. rhossiliensis showed the strongest parasitism and the best mortality according to their first comparative experiments (Fig. 3a)?
Minor points:
1. l. 70-71: this last sentence is read rather like a conclusion of the authors instead of a fact of the state of the art. Please revise.
2. Fig. 2: A larger magnification is needed for showing the attached conidial spores and the forming germinating tubes on the nematode cuticle and thus, differentiate them from the nematode's body autofluorescence, as the provided S3 Figure also verifies. The added photos a' - e' (and those of the supplement) help in resolving what the authors want to show, but if possible, photos of larger magnification would be better.
Comments on the Quality of English LanguageMinor point:
The authors must go through the text again and correct any minor mistakes like the one in l. 356 ("except" instead of "excepting")
